# Hepatitis C Virus Infection Associated with Oral Potentially Malignant Disorder, Oral Cancer, and Liver Diseases: A Community-Based Cross-Sectional Study

**DOI:** 10.3390/cancers17223695

**Published:** 2025-11-18

**Authors:** Donlagon Jumparway, Chen-Yang Hsu, Amy Ming-Fang Yen, Ting-Yu Lin, Saman Warnakulasuriya, Tony Hsiu-Hsi Chen, Dih-Ling Luh, Chiu-Wen Su, Pongdech Sarakarn, Yen-Po Yeh, Sam Li-Sheng Chen

**Affiliations:** 1School of Dentistry, College of Oral Medicine, Taipei Medical University, Taipei 110, Taiwan; d204110002@tmu.edu.tw; 2Department of Internal Medicine, Daichung Hospital, Miaoli 350, Taiwan; bacillihsu@ntu.edu.tw; 3Taiwan Association of Medical Screening, Taipei 103, Taiwan; 4Master of Public Health Degree Program, College of Public Health, National Taiwan University, Taipei 100, Taiwan; 5School of Oral Hygiene, College of Oral Medicine, Taipei Medical University, Taipei 110, Taiwan; amyyen@tmu.edu.tw; 6Institute of Health Data Analytics and Statistics, College of Public Health, National Taiwan University, Taipei 100, Taiwan; abbielin@ntu.edu.tw (T.-Y.L.); chenlin@ntu.edu.tw (T.H.-H.C.); 7Faculty of Dentistry, Oral & Craniofacial Sciences, King’s College London, London WC2R 2LS, UK; saman.warne@kcl.ac.uk; 8Department of Public Health, Chung Shan Medical University, Taichung 402306, Taiwan; luh@csmu.edu.tw; 9Department of Family and Community Medicine, Chung Shan Medical University Hospital, Taichung 402306, Taiwan; 10Department of Internal Medicine, College of Medicine, National Taiwan University, Taipei 100229, Taiwan; chiuwensu@ntu.edu.tw; 11Faculty of Public Health, Khon Kaen University, Khon Kaen 40002, Thailand; spongd@kku.ac.th; 12Changhua County Public Health Bureau, Changhua 500009, Taiwan; 13Graduate Institute of Clinical Medicine, College of Medicine, National Taiwan University, Taipei 100229, Taiwan; 14Institute of Epidemiology and Preventive Medicine, College of Public Health, National Taiwan University, Taipei 10031, Taiwan

**Keywords:** hepatitis C viremia, risk factor, oral potentially malignant disorder, oral cancer, HCV-related liver diseases

## Abstract

This community-based cross-sectional study assessed the association between Hepatitis C Virus (HCV) infection and Oral Potentially Malignant Disorders (OPMDs) and oral cancer in 37,720 Taiwanese adults. After adjusting for confounders, HCV viremia was significantly associated with an elevated risk of OPMDs but not oral cancer, possibly due to competing HCV-related liver diseases. S Traditional risk factors, including smoking, alcohol use, areca nut chewing, obesity, and metabolic syndrome, were confirmed as significant independent risks. The findings suggest that HCV infection may contribute to early oral carcinogenesis, highlighting the need to integrate HCV screening and antiviral therapy into oral cancer prevention strategies.

## 1. Introduction

Oral potentially malignant disorders (OPMDs) play a critical role in the natural progression of oral cancer. These disorders, which include leukoplakia, erythroplakia, erythroleukoplakia, oral submucous fibrosis, and lichen planus, often represent a surrogate in the transformation of normal tissue to malignancy [1]. Thus, better understanding the risk factors associated with OPMDs is helpful for secondary oral cancer prevention.

Owing to the carcinogenic properties and chronic inflammatory effects, lifestyle behaviors including smoking, alcohol consumption, and areca nut chewing have been identified as the major risk factors for the development of OPMDs [2,3,4,5]. Human papillomavirus (HPV) is also recognized as the contributing factor in the development of OPMDs [6].

However, other virally induced factors, such as Hepatitis C virus (HCV) infection, have received relatively limited attention, despite their potential role in modulating cancer risk. The link between HCV infection and lichen planus has been substantiated by previous studies [7,8,9]. Previous study indicates that patients with diabetes and HCV are more likely to develop severe tubular atrophy, interstitial inflammation, and fibrosis [10], reflecting a chronic systemic inflammatory state. Chronic inflammation plays a crucial role in the pathogenesis of both systemic and oral diseases [11]. Therefore, chronic HCV may also be more susceptible to causing oral mucosal inflammation and potentially malignant conditions such as oral submucous fibrosis.

The association between HCV infection and oral malignancies is worthy of being elucidated in areas and regions known to be endemic for HCV infection and with high prevalence of areca nut chewing. Chronic HCV infection is also associated with several disorders, including diabetes, chronic kidney disease, renal cell carcinoma, pancreatic cancer, non-Hodgkin’s lymphoma, biliary tract cancer, and hepatocellular carcinoma [12,13,14,15,16,17,18,19,20].

To throw light on the effects of HCV infection on OPMDs and oral cancer, while adjusting for the competing HCV-related fatal diseases, particularly liver cancer and cirrhosis, a large population-based study is required. A community-based screening program in Changhua, Taiwan, offered a valuable opportunity to explore these research questions based on a large population-based dataset [21]. Thus, our study aimed to investigate the association between HCV infection and the occurrence of OPMD and oral cancer while adjusting for HCV-related liver diseases using a cross-sectional design. Through an analysis of demographic, behavioral, and clinical characteristics, this study simultaneously examines these outcomes related to HCV and evaluates whether individuals with HCV infection have an elevated risk of OPMD and oral cancer.

## 2. Materials and Methods

### 2.1. Study Design and Population

The targeted population consisted of 92,322 participants aged 30 years or above in Changhua, Taiwan. Given that oral cancer incidence is rare in individuals below age 30, this age threshold was selected to align with the appropriate screening population and ensure clinical and public health relevance. The study participants were those invited to the community-based integrated screening program, who received screening for five neoplastic diseases (liver cancer, breast cancer, colorectal cancer, oral cancer, and cervical cancer) and nonneoplastic diseases (hyperlipidemia, hypertension, and hyperglycemia) during the periods between 2005 and 2014. Oral cancer screening was performed every two years through oral visual inspections, focusing on high-risk groups, including smokers or areca nut chewers. Because of integrated liver cancer screening, data on the liver functions, including HCV infection, were available [21]. Out of an initial 92,322 participants, 37,720 eligible individuals who had received oral visual inspections were included after excluding those without HCV or OPMD results, individuals under 30, and some with incomplete information. The flowchart of the study framework is illustrated in Figure 1, which consists of data collection based on the outcome of HCV infection.

### 2.2. Case Identification and Data Collection

OPMD cases were identified based on clinical diagnoses made by dentists, otolaryngologists, or trained physicians during oral visual inspections conducted as part of an oral cancer screening program in Taiwan. The clinical assessments followed established diagnostic criteria for recognizing potentially malignant lesions, including oral leukoplakia, oral erythroleukoplakia, oral erythroplakia, oral submucosa fibrosis, verrucous hyperplasia, and oral lichen planus. The distribution of OPMD subtypes was presented in Appendix A. This classification highlights that leukoplakia constituted the majority of OPMDs observed, followed by oral submucosa fibrosis, with a small proportion comprising other subtypes. Histopathological confirmation was specifically required for cases that presented with induration, ulceration, or non-homogeneous appearance suggestive of malignancy. Punch biopsy was performed to confirm oral cancer diagnoses in patients with clinically suspicious lesions [22]. In addition to screen-detected oral cancer, oral cancers were ascertained by the national cancer registry after follow-up. The status of cirrhosis or liver cancer was identified through a liver cancer screening program [21].

Detailed information was collected across multiple domains, including demographics, behaviors, and clinical characteristics, with a focus on the following key variables. (1) Demographic factors, including age, gender, and education, were recorded. Education levels were categorized as elementary school or below, junior high school, senior high school, college, or above. (2) Oral habits, specifically the history and frequency of smoking, alcohol consumption, and areca nut chewing, were classified as “never,” “former,” or “current.” (3) Body Mass Index (BMI) was categorized into non-obesity (<30 kg/m^2^) and obesity (≥30 kg/m^2^) (4) Metabolic syndrome was determined based on the presence of at least three criteria: abdominal obesity, elevated blood pressure, abnormal fasting glucose, low high-density lipoprotein (HDL) cholesterol, or high triglycerides. (5) Hepatitis C virus (HCV) carrier detected by third-generation anti-HCV test (GB NANBASE C-96 3.1, Hsinchu County, Taiwan). The cases with an HCV signal-to-cutoff (S/CO) ratio ≥ 1 were considered as HCV positives. Subjects with positive anti-HCV antibodies underwent an HCV RNA test, a reflex test for hepatitis C viremia. Among 1913 HCV-positive individuals, 513 subjects did not undergo the HCV RNA test. Since 2018, direct-acting antiviral (DAA) treatment has been provided to individuals with hepatitis C viremia through the Hepatitis C Virus Elimination Program [23].

### 2.3. Statistical Analysis

The multinominal logistic regression was used to examine the relationship between HCV infection and the outcomes of interest, including OPMD, oral cancer, and HCV-related liver diseases (cirrhosis or liver cancer). The analysis gave rise to adjusted odds ratios (aORs) and 95% confidence intervals (CIs), which quantified the associations while controlling for confounders including age, sex, education level, lifestyle behaviors, and metabolic syndrome. Both univariate and multivariable multinominal logistic regression analyses were performed to assess the role of HCV associated with OPMD and oral cancer while adjusting for HCV-related liver diseases. All analyses were performed using SAS software, version 9.4 (SAS Institute Inc., Cary, NC, USA).

## 3. Results

Among 37,720 eligible individuals, 4.8% (n = 1816), 0.89% (n = 336), and 0.59% (n = 223) were classified as having OPMD, oral cancer, and HCV-related liver disease (cirrhosis or liver cancer), respectively (Table 1). The prevalence of hepatitis C viremia was higher in OPMD cases (4.4%), slightly higher in patients diagnosed with oral cancers (3.3%), and extremely higher in patients diagnosed with cirrhosis or liver cancers (23.3%) compared to the screen negative group (2.7%) (Table 1).

The highest proportion of OPMD cases (40.8%) was noted in individuals aged 50–59, followed by those aged 40–49 (26.9%), 60–69 (24.9%), and the lowest in younger individuals (aged 30–39) (1.3%). The age distribution of oral cancer revealed a progressive disease with a natural pathway from OPMD to oral cancer with advancing age while also occurring alongside cirrhosis or liver cancer, particularly in HCV-positive subjects. Those susceptible to the pathway of OPMD-oral cancer would develop oral cancer if they had not died from cirrhosis or liver cancer.

A marked gender disparity was observed. Males constituted 96% of OPMD cases, or oral cancers, compared to only 4% females, emphasizing a predominance of the disease among men. Educational attainment also varied significantly between groups. A larger proportion of OPMD cases (40.8%) had received only elementary school education or below, compared to 7.4% who had attained college-level education. More than 50% of patients diagnosed with oral cancer and cirrhosis or liver cancer had only an elementary school education or below. This indicates a potential link between lower educational attainment and a higher risk of OPMD, oral cancer, and HCV-related liver disease. Behaviors such as smoking, alcohol consumption, and areca nut chewing were significantly more prevalent among individuals with OPMD, oral cancer, and HCV-related liver disease. Among these, 68.0% of OPMD cases, 52.1% of oral cancer cases, and 41.3% of HCV-related liver disease cases were current smokers, compared to 24.0% in the screen-negative group. Similarly, current alcohol drinkers accounted for more OPMD (32.0%), oral cancer (36.0%), and HCV-related liver disease (23.3%) compared to the disease-free group (12.9%). Areca nut chewing, a well-documented risk factor for oral lesions, was reported by 28.9% of current OPMD cases and 30.1% of former users. The prevalence of areca nut use (15.3%) was also higher among individuals with HCV-related liver diseases, whereas 81.2% of the screen-negative group had never been engaged in areca nut chewing. Obesity (BMI ≥ 30) was more prevalent among OPMD cases (11.7%), oral cancers (8.7%), and HCV-related liver diseases (9.0%) compared to the screen-negative group (7.1%). Similarly, metabolic syndrome was identified in 39.8% of OPMD cases and 35.5% of oral cancers, whereas only 29.4% of HCV-related diseases and 29.5% of the screen-negative group exhibited this condition.

In the univariate multinomial logistic regression model, several factors were found to be significantly associated with the risks of OPMD, oral cancer, and cirrhosis or liver cancer (Table 2). Hepatitis C viremia was strongly associated with an increased risk of cirrhosis or liver cancer and modestly associated with OPMD. Lower education levels increased odds of cirrhosis or liver cancer, while those with junior high education had elevated risks of OPMD and oral cancer. Current smoking was strongly associated with both OPMD and oral cancer. Alcohol consumption and areca nut chewing significantly increased risks for all three conditions. Obesity and metabolic syndrome were significantly associated with increased risk of OPMD, and metabolic syndrome also showed a significant association with oral cancer.

Table 3 shows individuals with hepatitis C viremia had a 50% statistically significantly higher risk of OPMD after adjustment for confounders (aOR = 1.50, 95% CI: 1.17–1.92), representing a moderate effect size. In contrast, no significant association was found between hepatitis C viremia and oral cancer (aOR: 1.09, 95% CI: 0.59–2.01), possibly attenuated by its strong association between hepatitis C viremia and cirrhosis/liver cancer (aOR = 11.59, 95% CI: 8.33–16.13), indicating a very large effect size.

Male (aOR = 4.68, 95% CI: 3.66–5.98), lower education levels (aOR from 1.45 to 1.64), former (aOR = 2.03, 95% CI: 1.68–2.44) or current smoker (aOR = 4.57, 95% CI: 3.91–5.34), current drinker (aOR = 1.15, 95% CI: 1.02–1.29), former (aOR = 1.82, 95% CI: 1.60–2.07) or current areca nut chewer (aOR = 2.62, 95% CI: 2.28–3.00), obesity (aOR = 1.51, 95% CI: 1.28–1.78), and metabolic syndrome (aOR = 1.36, 95% CI: 1.22–1.51) were associated with OPMD.

Male gender remained a significant risk factor for oral cancer (aOR = 4.28, 95% CI: 2.39–7.63). Compared to individuals with a college education, those with an elementary school education or below (aOR = 1.82, 95% CI: 1.12–2.95) or a junior high school education (aOR = 1.73, 95% CI: 1.06–2.81) had an increased risk of oral cancer. Current smoking continued to be a major risk factor for oral cancer (aOR = 1.57, 95% CI: 1.11–2.23). Similarly, former or current users of both areca nut and alcohol were significantly associated with increased risk of oral cancer. The adjusted odds ratios for all risk factors associated with OPMD and oral cancer are presented in Figure 2.

Male (aOR = 2.51, 95% CI: 1.65–3.82), lower education levels (aORs from 2.03 to 2.64), current drinker (aOR: 1.31, 95% CI: 0.90–1.91), and current areca nut chewer (aOR: 1.85, 95% CI: 1.17–2.95) were associated with cirrhosis or liver cancer.

## 4. Discussion

The present study explored the association between HCV infection and oral potentially malignant disorders (OPMDs) and oral cancer while controlling for the competing risk arising from HCV-related diseases in a large screening population. The findings provide valuable insights into the multifactorial etiology of OPMDs, demonstrating that HCV infection, along with traditional lifestyle factors such as smoking, alcohol consumption, and areca nut chewing, significantly contributes to the risk of potentially developing OPMDs and subsequent progression to oral cancer or liver diseases.

### 4.1. Association of HCV Infection with OPMDs and Oral Cancer

In addition to the established associations between HCV viremia and liver diseases, our new findings demonstrate that HCV viremia is independently associated with an elevated risk of OPMDs. This aligns with previous research emphasizing the role of HCV in systemic inflammation and its potential contribution to oral mucosal disorders, such as lichen planus [7,8,9]. Carrozzo et al. compiled evidence from meta-analyses, highlighting that HCV-associated inflammation may contribute to a microenvironment conducive to malignant transformation in oral tissues. The presence of HCV-specific T lymphocytes in the oral mucosa suggests a possible role of HCV in the pathogenesis of oral lichen planus [24]. Previous research on oral manifestations in HCV patients also provides valuable insights. Patients with HCV infection show more frequent changes in their oral mucosa, including erosion and angular cheilitis, compared to those without HCV [25,26]. Studies suggest that HCV may replicate in the oral mucosa, potentially contributing to mucosal damage [8,9]. HCV infection may play a significant role not only in affecting the oral mucous membrane but also in the development of oral cancer, with potential long-term impacts on oral health and an increased risk of malignant transformation over time. A meta-analysis conducted by Borsetto et al. examined the association between HCV infection and the risk of head and neck squamous-cell carcinoma (HNSCC). The analysis included eight studies from Taiwan, Japan, Australia, Italy, Denmark, and the USA and revealed significant risk correlations with specific cancer sites: oral cavity (relative risk (RR) = 2.13), oropharynx (RR = 1.81), and larynx (RR = 2.57). These findings highlight the need for monitoring of HCV-infected individuals for early detection of HNSCC and emphasize the importance of considering undiagnosed HCV in patients with HNSCC [27].

Despite previous studies elucidating the association between HCV and oral neoplasia, our study found that HCV infection was significantly associated with OPMD and HCV-related liver diseases but not oral cancer. Considering both past and present infections, most association studies have used HCV antibody status as an indicator of pathogenicity. Since HCV viremia reflects the pathogenic role of active infection in disease progression, it may better capture the active state of the virus and provide deeper insights into the role of HCV infection in disease development. We found a positive association between HCV viremia and OPMD (aOR = 1.50; 95% CI: 1.17–1.92), as well as HCV-related liver disease (aOR = 11.59; 95% CI: 8.33–16.13), but a slight increase in the risk of oral cancer (aOR = 1.09; 95% CI: 0.59–2.01) after adjusting for confounding factors (Table 3). Given the predominance of leukoplakia cases among the screen-positive group, we conducted a subgroup analysis on the association between hepatitis C viremia and leukoplakia, revealing an adjusted OR of 1.45 (95% CI: 1.08–1.95) (Appendix A). This finding is consistent with the primary results. However, due to the limited number of cases of other OPMD subtypes, the corresponding subgroup analyses did not yield significant results.

Our findings showed no association between HCV and oral cancer, which might contrast with previous research. Our community-based study examined the relationship between HCV and multiple medical conditions, including cirrhosis and liver cancer, as well as oral cancer. The timeline of risk factors in relation to the disease spectrum is illustrated in Figure 3. The longer development time for oral cancer may explain this discrepancy. Individuals with HCV typically develop liver diseases before oral cancer manifests and may die of liver disease, making the association less apparent. However, we were able to observe the impact of HCV on OPMD since OPMD occurs earlier than oral cancer.

### 4.2. Potential Biological Mechanisms and Interaction with Lifestyle Factors

Oral carcinogenesis might have the same virus-induced molecular mechanisms as hepatocarcinogenesis. HCV infection plays a crucial role in hepatocarcinogenesis through two pathways, including direct viral effects and indirect host response, leading to chronic liver injury, fibrosis, and malignant transformation. Chronic inflammation and oxidative stress are major drivers of HCV-related liver cancer development [28,29]. In addition to chronic inflammation, HCV would modulate molecular pathways that impair tumor suppressor mechanisms, specifically targeting the p53 pathway through the interaction of viral proteins, particularly NS5A [30,31], which intersects with pathways also implicated in oral cancer development. HCV infection also could induce epigenetic modifications and epithelial–mesenchymal transition (EMT), both of which contribute to fibrosis and increase the risk of HCC [28,30]. Those HCV-induced mechanisms might also contribute to oral carcinogenesis. Although a higher prevalence of HCV was observed in the OPMD and oral cancer groups, further longitudinal and mechanistic studies are needed to clarify the direct role of HCV in oral tumorigenesis.

In our findings, older adults had a lower risk of OPMD compared to younger age groups. However, the risk of oral cancer increased with age. This pattern underscores the unique age window of natural history of oral neoplasm, entering the early detectable OPMD from 40 years onwards, progressively growing between 50–59 years, and having malignant transformation to oral cancer or developing cirrhosis or liver cancer, depending on the status of HCV status. This phenomenon may also indicate that, without early detection through screening, OPMD in older individuals could progress to oral cancer over time. The incidence of OPMD and oral cancer was substantially lower in women than in men, which may largely reflect gender differences in lifestyle exposures. Consequently, further gender-stratified analyses were limited by small case numbers, particularly for oral cancer. Nonetheless, male gender remained a strong independent predictor after adjustment for other covariates. Lifestyle factors such as smoking, alcohol consumption, and areca nut chewing were identified as strong predictors of OPMDs, corroborating well-established literature on their carcinogenic and mutagenic effects on oral epithelial cells [1,3,4,32]. Notably, our findings also indicate that these factors may have synergistic effects when combined with HCV-related systemic changes, as highlighted by Worakhajit et al. in their study on OPMD risk factors across various populations [3]. While the precise mechanisms remain to be fully elucidated, plausible pathways include HCV-induced chronic inflammation and immune dysregulation interacting with known carcinogens such as tobacco, alcohol, and areca nut in the oral mucosa. These combined effects may impair mucosal repair, thereby promoting OPMD development and malignant progression.

Chronic HCV infection is known to worsen insulin resistance and metabolic syndrome, which may indirectly influence oral carcinogenesis [33,34]. We also found the association between metabolic syndrome and OPMDs, adding to the growing body of evidence linking metabolic disorders with cancer susceptibility. The interaction between systemic metabolic dysregulation and local oral risk factors requires further investigation, particularly in populations with a high prevalence of HCV. Although this study adjusted for multiple confounders, including age, sex, education, smoking, alcohol, areca nut use, obesity, and metabolic syndrome, the possibility of residual confounding cannot be excluded. Other unmeasured factors such as socioeconomic status, viral load, immune status, or human papillomavirus may also influence the observed association between HCV infection and OPMD. Therefore, further studies incorporating these parameters are needed.

### 4.3. Public Health Implications

The findings of this study have global relevance because HCV infection remains a widespread public-health burden. Our results may therefore have implications for populations in both high- and low-HCV-prevalence regions, supporting the need for multidisciplinary collaboration between hepatology and dental public-health programs. These findings not only highlight the importance of integrating HCV screening into routine health evaluations for populations at risk of OPMDs but also have significant implications for oral cancer prevention, potentially informing modified screening guidelines that include HCV infection as a factor for identifying high-risk individuals. In regions where both HCV and traditional lifestyle risk factors are prevalent, combined interventions targeting metabolic health and lifestyle modification could significantly reduce the burden of OPMDs. Successful HCV treatment is associated with not only beating liver cancer but also a reduced risk of oral cancer, underscoring the potential benefits of early and effective antiviral intervention in mitigating both systemic and oral health complications of HCV [35]. Early antiviral therapy for HCV, particularly with the advent of DAA therapy, may offer protective effects against the progression of oral cancer [26].

### 4.4. Limitations

We would wish to caution interpretation of our data due to a few limitations of the study. First, the limitation of histopathological confirmation being restricted solely to clinically suspicious lesions may have introduced a degree of diagnostic misclassification, particularly concerning the subtyping of OPMDs. Second, while an increased prevalence of HCV infection was noted among OPMD patients, this study focused on a high-risk population involved in oral cancer screening. Therefore, the findings might not be generalized to the general population. Although this study employed a cross-sectional design, limiting causal inference, it is worth noting that hepatitis C infection in Taiwan is generally persistent and long-standing, suggesting that the observed associations may reflect chronic rather than recent infection. Nevertheless, longitudinal follow-up studies are warranted to clarify the temporal relationship between HCV infection, OPMD onset, and subsequent malignant transformation.

## 5. Conclusions

In conclusion, this study highlights the HCV viremia associated with oral potentially malignant disorders (OPMDs) in a high-risk population, leveraging population-based screening data from the Changhua community in Taiwan. The findings demonstrate that HCV-positive status, coordinated with lifestyle factors such as smoking, alcohol consumption, and areca nut chewing, significantly increases the likelihood of detecting OPMDs. Additionally, systemic factors, including metabolic syndrome and obesity, further extend the risk, underscoring the complex interplay between behavioral, demographic, and metabolic determinants in OPMD development. These results underscore the value of integrated screening programs in identifying at-risk individuals and facilitating early interventions. By advancing our understanding of OPMD risk factors, this research offers actionable insights into public health policies to reduce the burden of OPMDs and prevent their progression to oral cancer.

## Figures and Tables

**Figure 1 cancers-17-03695-f001:**
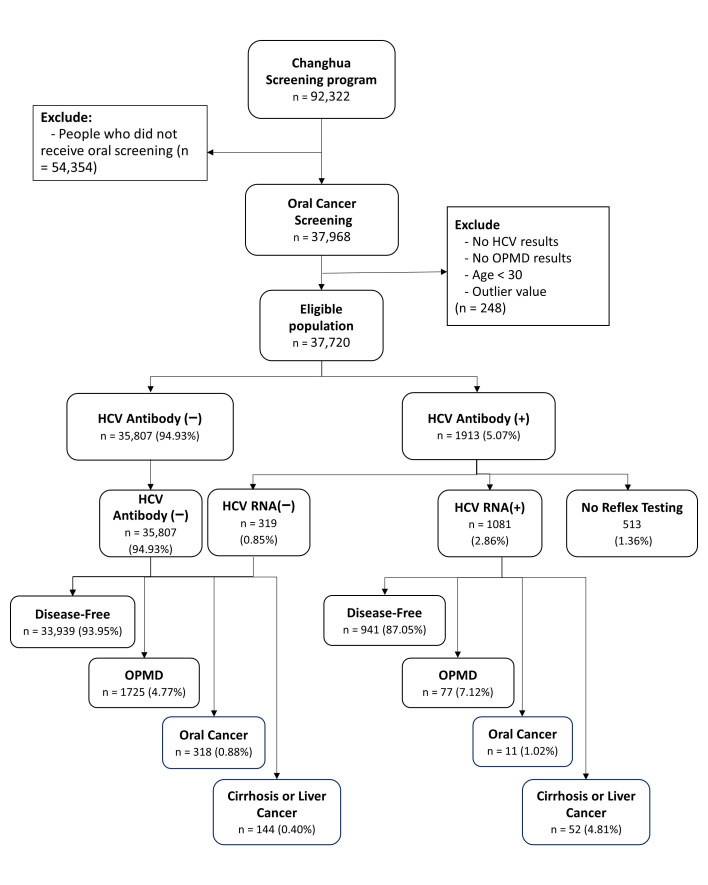
Flowchart Representing Data Collection in the Changhua Community-based Integrated Screening (CHCIS) Program.

**Figure 2 cancers-17-03695-f002:**
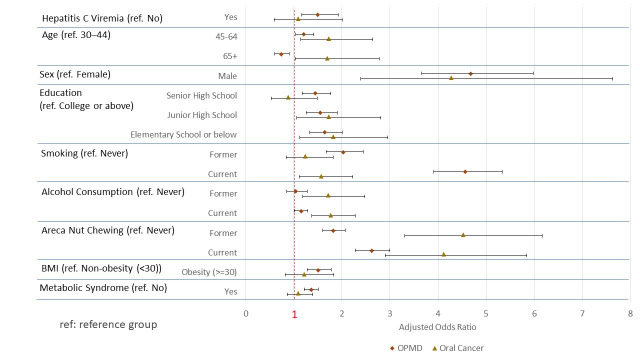
Adjusted odds ratios for factors associated with OPMD and oral cancer.

**Figure 3 cancers-17-03695-f003:**
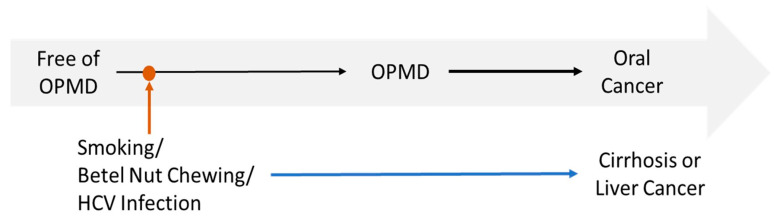
The Relationship of Risk Factors, Disease Progress, and Outcomes.

**Table 1 cancers-17-03695-t001:** Demographic Features of Disease-Free, OPMD, Oral Cancer, and Cirrhosis or Liver Cancer.

Variables	Screen-Negative		OPMD		Oral Cancer		Cirrhosis or Liver Cancer	
	n	%	n	%	n	%	n	%
**Total**	35,345	93.7%	1816	4.81%	336	0.89%	223	0.59%
**HCV**								
Negative	33,646	95.2%	1703	93.8%	318	94.6%	140	62.8%
Positive	1699	4.8%	113	6.2%	18	5.4%	83	37.2%
**Hepatitis C Viremia**								
No	34,380	97.3%	1737	95.6%	325	96.7%	171	76.7%
Yes	965	2.7%	79	4.4%	11	3.3%	52	23.3%
Not tested	465		14		7		27	
**Age** (Mean ± SD)	55.5 ± 11.1		55.0 ± 9.2		58.7 ± 9.4		57.3 ± 7.2	
**Age Group**								
30–39	9105	25.8%	24	1.3%	3	0.9%	1	0.5%
40–49	11,878	33.6%	488	26.9%	51	15.2%	26	11.7%
50–59	8682	24.6%	741	40.8%	125	37.2%	107	48.0%
60–69	3952	11.2%	452	24.9%	113	33.6%	84	37.7%
70+	9105	25.8%	111	6.1%	44	13.1%	5	2.2%
**Sex**								
Female	12,918	36.6%	78	4.3%	14	4.2%	40	17.9%
Male	22,427	63.5%	1738	95.7%	322	95.8%	183	82.1%
**Education**								
College or above	5324	15.1%	135	7.4%	21	6.3%	10	4.5%
Senior High School	8771	24.9%	472	26%	47	14%	46	20.6%
Junior High School	6484	18.4%	467	25.7%	89	26.6%	49	22%
Elementary School or below	14,666	41.6%	740	40.8%	178	53.1%	118	52.9%
**Smoking**								
Never	21,645	61.3%	266	14.7%	68	20.2%	102	45.7%
Former	5213	14.8%	315	17.4%	93	27.7%	29	13.0%
Current	8464	24.0%	1234	68.0%	175	52.1%	92	41.3%
**Alcohol Consumption**								
Never	29,424	83.3%	1110	61.2%	174	51.8%	156	70.0%
Former	1351	3.8%	125	6.9%	41	12.2%	15	6.7%
Current	4547	12.9%	580	32.0%	121	36.0%	52	23.3%
**Areca Nut Chewing**								
Never	28,696	81.2%	743	40.9%	100	29.8%	143	64.1%
Former	4357	12.3%	547	30.1%	157	46.7%	46	20.6%
Current	2269	6.4%	525	28.9%	79	23.5%	34	15.3%
**BMI**								
Non-obesity (<30)	32,735	92.9%	1603	88.4%	306	91.3%	202	91.0%
Obesity (>=30)	2510	7.1%	210	11.6%	29	8.7%	20	9.0%
**Metabolic Syndrome**								
No	24,734	70.5%	1087	60.3%	216	64.5%	156	70.6%
Yes	10,346	29.5%	716	39.7%	119	35.5%	65	29.4%

**Table 2 cancers-17-03695-t002:** Univariate Multinomial Logistic Regression Analysis of Factors Associated with OPMD, Oral Cancer, and Cirrhosis or Liver Cancer.

Variables	OPMD				Oral Cancer				Cirrhosis or Liver Cancer			
	aOR	95% CI		*p*-Value	aOR	95% CI		*p*-Value	aOR	95% CI		*p*-Value
**Hepatitis C Viremia**												
No	Ref.				Ref.				Ref.			
Yes	1.59	1.25	2.01	<0.001	1.22	0.67	2.23	0.521	12.73	9.21	17.60	<0.0001
**Age**												
30–44	Ref.				Ref.				Ref.			
45–64	1.41	1.24	1.62	<0.0001	2.44	1.63	3.64	<0.0001	6.11	3.00	12.43	<0.0001
65+	0.87	0.73	1.03	0.109	2.73	1.77	4.21	<0.0001	3.00	1.37	6.59	0.006
**Sex**												
Female	Ref.				Ref.				Ref.			
Male	12.74	10.14	16.00	<0.0001	12.91	7.56	22.07	<0.0001	2.74	1.89	3.96	<0.0001
**Education**												
College or above	Ref.				Ref.				Ref.			
Senior High School	2.14	1.76	2.60	<0.0001	1.36	0.81	2.28	0.240	2.55	1.28	5.09	0.008
Junior High School	2.86	2.35	3.47	<0.0001	3.50	2.17	5.64	<0.0001	3.71	1.87	7.37	<0.001
Elementary School or below	2.03	1.68	2.44	<0.0001	3.01	1.91	4.74	<0.0001	3.64	1.90	6.98	<0.0001
**Smoking**												
Never	Ref.				Ref.				Ref.			
Former	4.90	4.15	5.79	<0.0001	5.82	4.23	8.02	<0.0001	1.22	0.79	1.88	0.363
Current	11.89	10.38	13.61	<0.0001	6.75	5.07	9.00	<0.0001	2.12	1.56	2.88	<0.0001
**Alcohol Consumption**												
Never	Ref.				Ref.				Ref.			
Former	2.49	2.05	3.02	<0.0001	5.42	3.84	7.66	<0.0001	1.73	0.94	3.21	0.080
Current	3.37	3.03	3.74	<0.0001	4.60	3.63	5.83	<0.0001	2.03	1.44	2.85	<0.0001
**Areca Nut Chewing**												
Never	Ref.				Ref.				Ref.			
Former	4.86	4.33	5.46	<0.0001	10.22	7.91	13.20	<0.0001	2.06	1.43	2.95	<0.0001
Current	8.95	7.94	10.10	<0.0001	9.98	7.38	13.49	<0.0001	3.15	2.12	4.68	<0.0001
**BMI**												
Non-obesity (<30)	Ref.				Ref.				Ref.			
Obesity (>=30)	1.72	1.48	1.99	<0.0001	1.26	0.86	1.85	0.233	1.08	0.64	1.83	0.771
**Metabolic Syndrome**												
No	Ref.				Ref.				Ref.			
Yes	1.58	1.44	1.74	<0.0001	1.32	1.05	1.65	0.018	0.87	0.63	1.20	0.397

**Table 3 cancers-17-03695-t003:** Multiple Multinomial Logistic Regression Analysis of Factors Associated with OPMD, Oral Cancer, and Cirrhosis or Liver Cancer.

Variables	OPMD				Oral Cancer				Cirrhosis or Liver Cancer			
	aOR	95% CI		*p*-Value	aOR	95% CI		*p*-Value	aOR	95% CI		*p*-Value
**Hepatitis C Viremia**												
No	Ref.				Ref.				Ref.			
Yes	1.50	1.17	1.92	0.002	1.09	0.59	2.01	0.787	11.59	8.33	16.13	<0.0001
**Age**												
30–44	Ref.				Ref.				Ref.			
45–64	1.21	1.04	1.41	0.012	1.73	1.14	2.64	0.011	4.29	2.07	8.89	<0.0001
65+	0.74	0.60	0.91	0.004	1.70	1.03	2.78	0.036	1.73	0.75	4.00	0.198
**Sex**												
Female	Ref.				Ref.				Ref.			
Male	4.68	3.66	5.98	<0.0001	4.28	2.39	7.63	<0.0001	2.51	1.65	3.82	<0.0001
**Education**												
College or above	Ref.				Ref.				Ref.			
Senior High School	1.45	1.18	1.77	<0.001	0.88	0.53	1.49	0.642	2.03	1.01	4.08	0.046
Junior High School	1.55	1.26	1.90	<0.0001	1.73	1.06	2.81	0.029	2.31	1.15	4.64	0.019
Elementary School or below	1.64	1.33	2.01	<0.0001	1.82	1.12	2.95	0.015	2.64	1.35	5.18	0.005
**Smoking**												
Never	Ref.				Ref.				Ref.			
Former	2.03	1.68	2.44	<0.0001	1.24	0.84	1.82	0.276	0.66	0.40	1.09	0.102
Current	4.57	3.91	5.34	<0.0001	1.57	1.11	2.23	0.011	1.02	0.70	1.50	0.915
**Alcohol Consumption**												
Never	Ref.				Ref.				Ref.			
Former	1.04	0.85	1.28	0.694	1.72	1.19	2.48	0.004	1.23	0.64	2.38	0.537
Current	1.15	1.02	1.29	0.022	1.77	1.37	2.28	<0.0001	1.31	0.90	1.91	0.158
**Areca Nut Chewing**												
Never	Ref.				Ref.				Ref.			
Former	1.82	1.60	2.07	<0.0001	4.52	3.31	6.17	<0.0001	1.48	0.95	2.30	0.080
Current	2.62	2.28	3.00	<0.0001	4.12	2.90	5.85	<0.0001	1.85	1.17	2.95	0.009
**BMI**												
Non-obesity (<30)	Ref.				Ref.				Ref.			
Obesity (>=30)	1.51	1.28	1.78	<0.0001	1.22	0.82	1.83	0.328	1.26	0.72	2.18	0.419
**Metabolic Syndrome**												
No	Ref.				Ref.				Ref.			
Yes	1.36	1.22	1.51	<0.0001	1.09	0.86	1.39	0.470	0.77	0.55	1.08	0.128

## Data Availability

Data supporting this study are available from the corresponding author on reasonable request.

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
