# Peer review of "Hepatitis C Virus Infection Associated with Oral Potentially Malignant Disorder, Oral Cancer, and Liver Diseases: A Community-Based Cross-Sectional Study"

_cancers, 2025, doi:10.3390/cancers17223695_

Round 1
Reviewer 1 Report
Comments and Suggestions for Authors
Dear authors,
I would like to congratulate you on the quality of the work presented. However, there are a few minor issues to address:
1. Oral potentially malignant disorders (OPMDs) play a critical role in the natural pro- 56
gression of oral cancer. These disorders, which include leukoplakia, erythroplakia, 57
erythroleukoplakia, oral submucous fibrosis, and lichen planus, often represent a surro- 58
gate in the transformation of normal tissue to malignancy. Please add a reference.
2. In the Introduction section, please add references to support each statement.
3. In the Introduction, please include information that HCV is linked to OLP, which is actually an OPMD.
Ref. Di Stasio, D., Lucchese, A., Romano, A. et al. The clinical impact of direct-acting antiviral treatment on patients affected by hepatitis C virus-related oral lichen planus: a cohort study. Clin Oral Invest 26, 5409–5417 (2022). https://doi.org/10.1007/s00784-022-04507-9
4. Please add some graphics to better illustrate the important results.
5. In the title and the introduction section, please include the study design.
Author Response
Reviewer I
Dear authors,
I would like to congratulate you on the quality of the work presented. However, there are a few minor issues to address:
Q1. Oral potentially malignant disorders (OPMDs) play a critical role in the natural progression of oral cancer. These disorders, which include leukoplakia, erythroplakia, erythroleukoplakia, oral submucous fibrosis, and lichen planus, often represent a surrogate in the transformation of normal tissue to malignancy. Please add a reference.
Ans: Agreed. The reference has been added.(Page 2, Line 69)
Q2. In the Introduction section, please add references to support each statement.
Ans: Thankful reviewer’s valuable suggestion. The necessary references have been added. Reference 1, Reference 2-5, Reference 6, Reference 7-9, Reference 10, Reference 11, Reference 12-20, and reference 21 have been revised and added to support each statement.
Q3. In the Introduction, please include information that HCV is linked to OLP, which is actually an OPMD.
Ans: Information about the link between HCV and lichen planus has been included in the introduction as follows: “The link between HCV infection and lichen planus has been substantiated by previous studies [7–9].”(Page 2, Line 79-80). The suggested reference of Stasio’s study has been included in the discussion as follows “Early antiviral therapy for HCV, particularly with the advent of DAA therapy, may offer protective effects against the progression of oral cancer [26]” (Page 12, Line 369)
Q4. Please add some graphics to better illustrate the important results.
Ans: Thank you for your valuable comment. We have added the new Figure 2 to highlight the important findings.
Q5. In the title and the introduction section, please include the study design.
Ans: The study design has been included in the title and the introduction section.
Reviewer 2 Report
Comments and Suggestions for Authors
The study investigates the association between Hepatitis C virus (HCV) infection and the risk of oral potentially malignant disorders (OPMD), oral cancer, and liver diseases. Data from a large cohort of 37,720 participants revealed that HCV infection was associated with an increased risk of OPMD but not oral cancer. The study also emphasized the role of traditional risk factors such as smoking, alcohol consumption, and areca nut chewing in the development of oral diseases. The manuscript may be further improved by following suggestions.
- The word already mentioned in the title should not be repeated in keyword, select correct keyword relevant to the study.
- Use effect size for all statistical analyses to quantify the magnitude of differences or relationships, providing more meaningful insights beyond mere statistical significance.
- The use of a large population sample strengthens the validity of the findings, but the cross-sectional design limits causal inferences.
- The high prevalence of HCV in the OPMD and oral cancer groups warrants further investigation into its direct role in oral carcinogenesis.
- The association between HCV and OPMD may be influenced by other underlying factors not fully accounted for in the analysis.
- The study lacks a deeper exploration of gender-specific risk factors.
- The study would benefit from a more detailed examination of the effects of HCV viremia on the progression of OPMD and its potential to develop into oral cancer.
- The findings suggest a potential synergistic effect between HCV and lifestyle risk factors, yet the mechanism behind this interaction remains unclear.
- The lack of a longitudinal follow-up limits the ability to assess how HCV infection progresses to more severe stages such as liver cancer or oral cancer.
- Although the study controls for several confounders, factors such as socioeconomic status and access to healthcare could influence the results and should be considered.
- The results suggest that early antiviral treatment for HCV may offer protection against oral cancer, which is a promising avenue for future research in preventive medicine.
- How the findings of the present study are relevant on a global scale.
- Provide limitation of the study under separate heading may be after discussion part.
May be improved.
Author Response
Reviewer II
The study investigates the association between Hepatitis C virus (HCV) infection and the risk of oral potentially malignant disorders (OPMD), oral cancer, and liver diseases. Data from a large cohort of 37,720 participants revealed that HCV infection was associated with an increased risk of OPMD but not oral cancer. The study also emphasized the role of traditional risk factors such as smoking, alcohol consumption, and areca nut chewing in the development of oral diseases. The manuscript may be further improved by following suggestions.
Q1: The word already mentioned in the title should not be repeated in keyword, select correct keyword relevant to the study.
Ans: Thank you for reviewer’s comment. The keywords have been corrected. (Page 2, Line 62)
Q2:Use effect size for all statistical analyses to quantify the magnitude of differences or relationships, providing more meaningful insights beyond mere statistical significance.
Ans: Thank you for the suggestion. Effect sizes have now been clearly reported and interpreted in the Results section, using adjusted odds ratios with 95% confidence intervals to represent the magnitude of associations beyond statistical significance. (Page 7, Line 218-223)
Q3:The use of a large population sample strengthens the validity of the findings, but the cross-sectional design limits causal inferences.
Ans: This point has been addressed in the limitations. (Page 12, Line 377-382)
Q4:The high prevalence of HCV in the OPMD and oral cancer groups warrants further investigation into its direct role in oral carcinogenesis.
Ans: We appreciate the insightful comment. In the revised manuscript, we have added a statement as follows “Although a higher prevalence of HCV was observed in the OPMD and oral cancer groups, further longitudinal and mechanistic studies are needed to clarify the direct role of HCV in oral tumorigenesis.”. The relevant text has been added to the Discussion section (Page 11, Line 317-319).
Q5:The association between HCV and OPMD may be influenced by other underlying factors not fully accounted for in the analysis.
Ans: Thank you for this valuable suggestion. We have added a statement in the Discussion acknowledging that the association between HCV and OPMD could be affected by residual confounding from unmeasured factors as follows “Although this study adjusted for multiple confounders including age, sex, education, smoking, alcohol, areca nut use, obesity, and metabolic syndrome, the possibility of residual confounding cannot be excluded. Other unmeasured factors such as socioeconomic status, viral load, immune status, or human papillomavirus may also influence the observed association between HCV infection and OPMD. Therefore, the further studies incorporating these parameters are needed.” (Page 11, Line 347-353)
Q6:The study lacks a deeper exploration of gender-specific risk factors.
Ans: We appreciate the reviewer’s valuable comment regarding gender-specific risk factors. In this community-based cohort, the incidence of OPMD and oral cancer was substantially lower among women than men, largely reflecting gender differences in exposure to major behavioral risk factors such as smoking and areca nut chewing. Since female participants had lower rates of these behaviors, further gender-stratified analyses were limited by small case numbers, particularly for OPMD and oral cancer outcomes. We have clarified this in the Discussion as follows “The incidence of OPMD and oral cancer was substantially lower in women than in men, which may largely reflect gender differences in lifestyle exposures. Consequently, further gender-stratified analyses were limited by small case numbers, particularly for oral cancer. Nonetheless, male gender remained a strong independent predictor after adjustment for other covariates.” (Page 11, Line 327-331)
Q7: The study would benefit from a more detailed examination of the effects of HCV viremia on the progression of OPMD and its potential to develop into oral cancer.
Ans: Thank you for this valuable suggestion. As this was a cross-sectional study, disease progression could not be directly evaluated. We have acknowledged this limitation and noted in the Discussion that future longitudinal studies are needed to explore the effect of HCV viremia on OPMD progression and malignant transformation. (Page 12, Line 377-382)
Q8:The findings suggest a potential synergistic effect between HCV and lifestyle risk factors, yet the mechanism behind this interaction remains unclear.
Ans: We agree that the underlying mechanisms mediating the potential synergistic effects between HCV infection and lifestyle risk factors remain to be clarified. We have expanded the Discussion to note the possible explanations as follows “While the precise mechanisms remain to be fully elucidated, plausible pathways include HCV-induced chronic inflammation and immune dysregulation interacting with known carcinogens such as tobacco, alcohol and areca nut in the oral mucosa. These combined effects may impair mucosal repair, thereby promoting OPMD development and malignant progression.” (Page11, Line 336-341)
Q9:The lack of a longitudinal follow-up limits the ability to assess how HCV infection progresses to more severe stages such as liver cancer or oral cancer.
Ans: We fully agree that the lack of longitudinal follow-up limits our ability to assess disease progression and temporal relationships between HCV infection. We have acknowledged this limitation and noted in the Discussion that future longitudinal studies are needed. (Page 12, Line 377-382)
Q10:Although the study controls for several confounders, factors such as socioeconomic status and access to healthcare could influence the results and should be considered.
Ans: Agreed. We have added a statement in the Discussion acknowledging that the association between HCV and OPMD could be affected by residual confounding from unmeasured factors as follows “Although this study adjusted for multiple confounders including age, sex, education, smoking, alcohol, areca nut use, obesity, and metabolic syndrome, the possibility of residual confounding cannot be excluded. Other unmeasured factors such as socioeconomic status, viral load, immune status, or human papillomavirus may also influence the observed association between HCV infection and OPMD. Therefore, the further studies incorporating these parameters are needed.” (Page 11, Line 347-353).
Q11:The results suggest that early antiviral treatment for HCV may offer protection against oral cancer, which is a promising avenue for future research in preventive medicine.
Ans: Thank you for your comments.
Q12:How the findings of the present study are relevant on a global scale.
Ans: We thank the reviewer for this thoughtful comment. We have added a paragraph in the Discussion highlighting the global relevance of our findings as follows “The findings of this study have global relevance because HCV infection remains a widespread public-health burden. Our results may therefore have implications for populations in both high- and low-HCV-prevalence regions, supporting the need for multidisciplinary collaboration between hepatology and dental public-health programs.” (Page11, Line 355-358).
Q13:Provide limitation of the study under separate heading may be after discussion part.
Ans: Agreed. The subheading has been added in the Discussion.
Reviewer 3 Report
Comments and Suggestions for Authors
The study aims to explore any association of HCV infection with oral potentially malignant disorder (OPMD) and oral cancer, following adjustment for HCV-related liver disease in a large screening cohort. The study is interesting, and the manuscript is well-written and well-structured. There are some observed issues.
- It is not clear why the authors determined the age of 30 years or above as an inclusion criterion; why not less than 30 years? Please describe the motivation for determining this criterion in the methods section.
- For the diagnosis of OPMDs, the authors stated that the diagnosis is based on clinical visual inspections. This is inappropriate because the diagnosis of many OPMDs should be based on histological evaluations, and some of them are a differential diagnosis of others. Please describe better how the diagnosis was achieved for each OPMD.
- The authors did not mention any patient with oral lichenoid lesions (OLL); were there OLL cases included? This is interesting for this topic of the study because some reports describe a possible association between OLL and even oral lichen planus (OLP) and HCV. Please revise this issue in the study population and report it in the methods and results section, and discuss this issue in the discussion section.
- The authors did not report the calculated p-values for all the reported statistical results. Please report them even if they are not significant.
Author Response
Reviewer III
The study aims to explore any association of HCV infection with oral potentially malignant disorder (OPMD) and oral cancer, following adjustment for HCV-related liver disease in a large screening cohort. The study is interesting, and the manuscript is well-written and well-structured. There are some observed issues.
Q1. It is not clear why the authors determined the age of 30 years or above as an inclusion criterion; why not less than 30 years? Please describe the motivation for determining this criterion in the methods section.
Ans: We thank the reviewer for this important comment. The inclusion criterion of participants aged 30 years and above was based on the national oral cancer screening policy in Taiwan, where screening is offered to adults aged 30 years or above with risk behaviors such as smoking or areca nut chewing. Epidemiological data from Taiwan show that the incidence of oral cancer is extremely low in individuals younger than 30 years. Therefore, including this younger age group would contribute minimally to case detection. We have added this rationale in the Methods section (Page 3, Line 104-108).
Q2. For the diagnosis of OPMDs, the authors stated that the diagnosis is based on clinical visual inspections. This is inappropriate because the diagnosis of many OPMDs should be based on histological evaluations, and some of them are a differential diagnosis of others. Please describe better how the diagnosis was achieved for each OPMD.
Ans: We appreciate the reviewer for the helpful comment. We have clarified in Section 2.2 (Case Identification and Data Collection) that OPMD diagnoses were made clinically by trained dentists or physicians following standardized criteria, with punch biopsy and histological confirmation performed for clinically suspicious lesions (e.g., induration, ulceration, or non-homogeneous appearance). We have also acknowledged in the Discussion that histopathological confirmation was limited to such cases, representing a potential limitation of this study.
2.2. Case identification and data collection
OPMD cases were identified based on clinical diagnoses made by dentists, otolaryngologists or trained physicians during standardized oral visual inspections. These diagnoses were performed following clinical criteria for recognizing potentially malignant lesions, such as oral leukoplakia, oral erythroleukoplakia, oral erythroplakia, oral submucosa fibrosis, verrucous hyperplasia, and oral lichen planus. The distribution of OPMD subtypes was presented in Supplemental Table 1. This classification highlights that leukoplakia constituted the majority of OPMDs observed, followed by oral submucosa fibrosis, with a small proportion comprising other subtypes. Punch biopsy was performed to confirm oral cancer diagnoses in patients with clinically suspicious lesions. In addition to screen-detected oral cancer, oral cancers were ascertained by national cancer registry after follow-up. (Page 4, Line 123-136)
Q3. The authors did not mention any patient with oral lichenoid lesions (OLL); were there OLL cases included? This is interesting for this topic of the study because some reports describe a possible association between OLL and even oral lichen planus (OLP) and HCV. Please revise this issue in the study population and report it in the methods and results section, and discuss this issue in the discussion section.
Ans: Thank you for this insightful comment. Oral lichen planus (OLP) was indeed included as one of the recognized subtypes of oral potentially malignant disorders in this study. As shown in Supplemental Table 1, the OPMD category encompassed several clinically diagnosed lesions, including leukoplakia, oral submucous fibrosis, erythroleukoplakia, verrucous hyperplasia, erythroplakia, and oral lichen planus. Because of the limited number of OLP cases (n = 10; 0.6% of all OPMDs), further subtype-specific statistical analyses were not feasible. However, we agreed the potential relationship between HCV infection and OLP has also been acknowledged in the Discussion with supporting references.
Supplemental Table 1: The Distribution of Oral Potential Malignant Disorder Subtypes among Participants
|
Classification |
Number |
% |
|
Leukoplakia |
1287 |
70.9 |
|
Oral Submucous Fibrosis |
285 |
15.7 |
|
Erythroleukoplakia |
39 |
2.1 |
|
Verrucous Hyperplasia |
39 |
2.1 |
|
Erythroplakia |
32 |
1.8 |
|
Lichen Planus |
10 |
0.6 |
|
Other Suspect OPMDs |
124 |
6.8 |
|
Total |
1,816 |
100.0% |
Q4. The authors did not report the calculated p-values for all the reported statistical results. Please report them even if they are not significant.
Ans: The P-value has been included in all tables.
Round 2
Reviewer 2 Report
Comments and Suggestions for Authors
Accepted
Comments on the Quality of English LanguageMay be improved.
Reviewer 3 Report
Comments and Suggestions for Authors
The manuscript has been improved after the authors' revision